# Text-Guided Data Attribution: Attributing the Influence of Simplicity Bias to Dataset

## Abstract

The effectiveness of deep learning models heavily relies on the quality and diversity of their training data. However, datasets collected from different sources often introduce simplicity biases, where a models rely on easily learnable but non-predictive (spurious) features for its predictions. While existing debiasing techniques focus on model robustness, they leave the data untouched. Further, they require manual group annotation of the entire training data or changes in training strategy, which are often constrained by privacy, regulatory, or proprietary constraints. As data becomes increasingly valuable, identifying and mitigating bias directly at the data level has gained importance. Recently, data attribution has emerged as a promising tool for uncovering issues in training data, yet its vulnerability to simplicity bias has received limited attention. In this work, we propose a novel data deletion framework that combines Neural Tangent Kernel (NTK)-based data attribution with textual descriptions of bias to identify and remove training samples that do not significantly affect model performance. We first demonstrate that NTK-based data attribution methods can themselves be influenced by spurious features. Subsequently, to mitigate this, we use available metadata or, when unavailable, a vision-language model, to annotate a small validation set and extract a textual description of the bias. Based on this description, we identify training samples that are semantically aligned with the spurious feature and exhibit high detrimental attribution scores. Removing these samples from the training data and retraining the model on the new training set improves its performance. Our approach achieves better average and worst-group accuracy, outperforming existing attribution-based baselines.

## 1 Introduction

The success of deep learning models is strongly influenced by the quality and quantity of the dataset used for training [1–4]. These data are often collected via web scraping [5, 6], and external data providers [7, 8]. However, such datasets can inadvertently contain illegal content [9] and can encode negative societal biases [10, 11] that can influence model performance. In addition, data collected from such varied sources can introduce distributional shifts, where subpopulations with specific features may be overrepresented or underrepresented in the training data compared to the test data [12].

These imbalances can introduce simplicity bias [13–15] where the model, due to high correlations between specific features and the prediction task, relies on simpler, non-robust (spurious) features instead of learning predictive features for classification. Several methods have been proposed to handle such biases in the model. However, instead of addressing the data as the fundamental source of bias, they primarily focus on improving model robustness by reweighting the loss function [16], modifying

Submitted to 39th Conference on Neural Information Processing Systems (NeurIPS 2025). Do not distribute.

the training objective [17–19], and model fine-tuning [20]. While effective, they inadvertently modify the standard training pipeline, which can increase the model's susceptibility to adversarial attacks [21–24], and could conflict with regulatory requirements, especially in safety-critical settings [25, 26], which require adherence to a specific training regime for theoretical guarantees. Further, considering the inherent proprietary value of data [27] and the monetary investment needed for collecting a new dataset, it has become increasingly important to address these challenges at the data level.

A viable alternative in these scenarios could be to remove training samples containing spurious features [28, 29, 11], by ensuring that these samples don't hurt the overall performance of the model, as in data attribution and Leave-One-Out (LOO) techniques [30–32]. Data attribution methods aim to estimate a model's performance when specific training samples are excluded, enabling the evaluation of counterfactual scenarios—such as assessing the impact on test accuracy if certain subsets of the training data were omitted [31, 33, 34]. However, many of these methods are computationally expensive and can underperform in non-convex settings. Recent advancements in data attribution methods, such as Trak [32], leverage neural tangent kernels (NTK) to enable scalable data attribution for non-convex models [32]. However, the impact of spurious features on the data attribution scores generated by such methods remains an open question.

In our work, we demonstrate (Proposition 1, Appendix J) that in the presence of data bias, methods like Trak [32] can undervalue the attribution scores for training samples with spurious features [13–15]. This misattribution can hinder the identification of detrimental samples, especially for methods that rely solely on the magnitude of attribution scores [35, 31].

Motivated by these observations, we propose a two-stage strategy to mitigate the impact of spurious features - (a) In the first stage, we focus on identifying such features within the dataset using available meta-data or annotations generated by a vision language model. (b) In the second stage, we use multimodal embeddings, such as CLIP [36] to learn a metric [37, 38] that identifies training examples that are semantically similar to the spurious features identified in the first step and whose removal can improve the model's performance as per the attribution scores.

The spurious features in the first stage are identified using metadata wherever available. In cases where metadata is unavailable, we utilize a vision-language model (VLM) to annotate a small validation set with its respective attributes and their associated values that are likely to introduce simplicity biases [39–41]. By evaluating the model's performance on these attribute-value pairs and comparing it to the overall performance on the validation dataset [42], we identify potential spurious features and generate a corresponding textual description of these biases [43]. This textual representation enables targeted data pruning and helps to mitigate the impact of spurious features without relying on manual group annotations in the training dataset.

In summary, our contributions in this paper are as follows:

- We propose a novel data-centric approach that combines NTK-based data attribution methods with textual descriptions of underlying bias to mitigate the impact of spurious features in training datasets.
- We first theoretically demonstrate that NTK-based attribution scores can be influenced by spurious features, which may limit the effectiveness of methods that rely solely on these scores for data pruning. To overcome this limitation, we introduce a metric learning-based data deletion strategy that selectively removes training samples aligned with textual descriptions of spurious features and exhibiting low attribution scores.
- Our approach achieves up to a 4% gain in average accuracy, 18% in worst-group accuracy, and a 50% improvement in class-level performance across various datasets. Additionally, it outperforms NTK-based methods like Trak on average by 10.6% in worst-group accuracy for different biased datasets.

## 2 Related Work

### 2.1 Data Attribution

Data attribution methods provide a framework to relate a model's predictions to its training dataset and have been used in a wide range of tasks, including model debugging and repair [44–47], subset selection [33, 34, 48], group robustness [11] and removing poisoning attacks [49].

The idea of linking a model's predictions to its training data has been studied for decades under various names, including influence functions [50], regression analysis [51], and jackknife methods [52]. However, most of these early works focused on linear models and aimed to predict changes in the optimal parameters when individual or groups of samples were excluded during the learning process. Recent works have tried to extend influence function and jacknife-based attribution methods to non-linear models and bigger datasets [30, 53, 54]. However, despite their promising predictive capabilities, these methods often make strong assumptions of strong convexity and the existence of a unique global solution, which are not applicable for neural networks [55]. Furthermore, Basu et al. [56], Hammoudeh and Lowd [57] have demonstrated the fragile nature of methods like influence functions across different architectures, showing that they sometimes fail basic sanity checks. Various approaches have been proposed to address the limitations of influence functions, including gradient agreement scoring [58], training models to predict attribution scores, as in DataModels [59], and methods like Trak [32], which leverage concepts from the Neural Tangent Kernel (NTK) for data attribution. Unlike other approaches, such as DataModels, Trak does not require training thousands of models [32, 59] or tracking the loss changes over the entire training process, making it more efficient. However, the impact of spurious features within the dataset on the data attribution method like Trak remains largely unexplored.

## 2.2   Spurious Features and Simplicity Bias

Spurious features often arise from selection bias in the dataset [60], where, in the presence of multiple hypotheses for prediction, the model tends to rely on the simplest feature [61, 14, 13]. This preference can lead to suboptimal model performance, as it often ignores more robust and meaningful features that are essential for generalization in real-world scenarios. Various methods have been proposed to address spurious features in models. These include data augmentation techniques [62–67], and learning strategies that change the training objectives to make the model robust to spurious features [17, 68, 69, 19, 16, 70, 20]. However, many of these changes are restricted under the regulatory policy for safety-critical applications [71–74], especially considering privacy concerns associated with collecting datasets and model certification-based requirements [75, 76]. Recent work has explored data deletion as a strategy for mitigating spurious features [28, 29, 11]. These methods use group annotation of the dataset to remove random samples from majority groups [28, 29] or those with high detrimental attribution scores [11]. However, these methods often require manual group annotation of training [28, 29] or validation data [11], which is costly and time-consuming. Further, in real-world settings, where biases are identified post hoc after deployment and evolve over time [77], generating such annotations is often impractical, and enforcing a balance among different groups may result in excessive data removal from the majority group and can harm generalization [29]. Our method circumvents these limitations by using text-guided data attribution to efficiently remove harmful samples within a deletion budget, without relying on group labels or hurting model performance. Further details on limitations and capabilities of existing methods are discussed in Appendix E.

## 3   Proposed Method

### 3.1   Problem Definition

Consider a classification setting with a training dataset $\mathcal{D}_{\text{train}} = \{z_1, \ldots, z_n\}$, where each sample $z_i = (x_i, y_i)$, consisting of an input $(x_i)$ and associated class label $(y_i)$ and a validation dataset, $\mathcal{D}_{\text{val}} = \{v_1, \ldots, v_m\}$ with validation samples $v_j = (x_j, y_j)$. The training dataset $(\mathcal{D}_{\text{train}})$ is used to train a neural network with optimal parameters $\theta^*(\mathcal{D}_{\text{train}})$. Additionally, we assume that $|\mathcal{D}_{\text{val}}| \ll |\mathcal{D}_{\text{train}}|$.

Suppose for every training sample $z$ there exists $t$ underlying hidden discrete attributes, $A' = \{a^1, \ldots, a^t\}$ and for each attribute $(a^j)$ there are $o$ possible values denoted as $V(a^j) \in \{b_1^j \ldots b_o^j\}$. In real-world settings, neural networks $(\theta^*)$ trained on $\mathcal{D}_{\text{train}}$ often associate class labels $(y)$ with specific attribute-value pairs $(a^m, b_t^m)$ [43, 13, 14]. For example, a model trained to predict gender might associate it with the feature "beard" (present/absent). However, feature imbalance in the datasets can lead to misleading associations. If most of the male images in a dataset include smiles, the model might spuriously link "male" with "smiling" rather than "beard." This can cause misclassification,

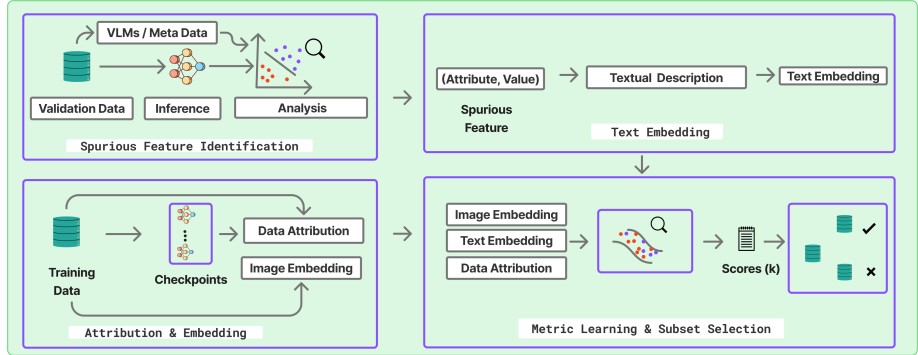

Figure 1: The figure illustrates the key steps in identifying detrimental samples. First, the performance of the model across different attribute value pairs is analyzed to identify and textually describe the underlying bias. Then, training samples that align with this bias and exhibit high detrimental attribution scores are selected for removal.

like predicting smiling females as males. We term such misleading attribute-value pairs as **spurious features**. In these scenarios, the primary objective of our work is to identify a set of detrimental examples, $\mathcal{S}^{\text{deter}} \in \mathcal{D}_{train}$, that "correspond" to the spurious features and might degrade the model's performance. The model is then retrained from scratch on the filtered dataset, $\left(\mathcal{D}_{\text{train}} \setminus \mathcal{S}^{\text{deter}}\right)$, to reduce the influence of the spurious feature in the training dataset similar to prior work like Chaudhuri et al. [28].

Our method for identifying $\mathcal{S}^{\text{deter}}$ involves two steps: (1) Annotate attribute–value pairs in the validation set to detect potential spurious features and generate a textual description of the bias; (2) Select $\mathcal{S}^{\text{deter}} \subset \mathcal{D}_{\text{train}}$ as samples semantically aligned with the bias and whose removal as per the data attribution scores does not degrade model performance.

### 3.2 Attribute Annotation and Spurious Feature Identification

A key component to identify spurious features is the availability of attribute–value annotations for the validation dataset. However, in many practical scenarios, such annotations are often missing from the metadata. Chen et al. [39] has shown that in the absence of such information, large language and vision models can be used to generate annotations necessary to identify the underlying spurious features. Hence, for datasets without pre-annotated attributes, we annotate the validation set with potential attribute–value pairs to assist in identifying spurious features.

To generate candidate attribute–value pairs, we leverage large language models such as ChatGPT [39]. ChatGPT is provided with a simple task description and prompted to suggest relevant attributes and associated values. For example, for a gender classification task, it can generate attributes like "smile", "beard", with possible values as "presence" or "absence". We adopt task-specific prompts proposed by Chen et al. [39] to guide this process. Once the attribute–value pairs are generated, the next step is to annotate the validation dataset. However, considering the limitations associated with ChatGPT for this task [39], we use Llama 3.2 [78], a vision–language model, to annotate the images in the validation dataset. Further details about the prompts can be found in Appendix G.

### 3.2.1 Spurious Feature Identification

To identify spurious features, we take motivation from recent work that tries to identify systematic bias in a model [42, 43] based on its accuracy and errors on the validation dataset. However, unlike previous methods, which try to identify underperforming subgroups that may require collecting additional data, we try to determine the overperforming attribute-value pair as a possible candidate for data deletion [79, 28]. For this, we take inspiration from Johnson et al. [42] and compare the performance of the dataset associated with each attribute-value pair to the performance of the entire dataset. If the performance gap exceeds a predefined threshold, the corresponding attribute-value pair is flagged as a potential spurious feature in the model. Formally, this is expressed as:

$$\frac{1}{|\mathcal{D}_\alpha|} \sum_{(x,y)\in\mathcal{D}_\alpha} \mathbf{1}\big(h(x)=y\big) - \frac{1}{|\mathcal{D}_{\text{val}}|} \sum_{(x,y)\in\mathcal{D}_{\text{val}}} \mathbf{1}\big(h(x)=y\big) > \tau, \tag{1}$$

where, $\mathcal{D}_\alpha$ is a subset of validation data $\mathcal{D}_{val}$ associated with $a^v$ attribute and it's $j^{th}$ value $b_j^v$. The indicator function $\mathbf{1}$ indicates the correct prediction made by the model. The function $h(x)$ represents the prediction made by the model for a given input x, and y as the corresponding true class label. The parameter $\tau$ denotes the minimum threshold.

Once an attribute-value pair exceeds the threshold, a textual description is generated to describe the spurious feature. For example: *"Images with {a} as {b}."* Here, $(a, b)$ is the attribute value pair selected as per Equation 1. Details about the textual description are provided in Appendix H.4.

## 3.3  Coherent Data Attribution

After generating the desired text, the next task is to select a subset of data that is semantically coherent with the given text and whose removal can improve the performance of the model [80].

Since our task involves efficient subset selection, we formally define data attribution as follows:

**Definition 1** (Data Attribution and Leave-one-out Influence Score [32])**.** *Given training dataset $\mathcal{D}_{train}$, and a model's utility function $\boldsymbol{f}(v;\theta)$ that measures the performance of the model, the data attribution score $\alpha : \mathcal{D}_{train} \times \mathcal{D}_{val} \rightarrow \mathcal{R}$ is defined as the change in the model's prediction for a validation sample $v_i$ with respect to the optimal parameters when the training example $z_k$ is excluded from the training dataset during the learning of the optimal parameters $\theta^*$. Formally,*

$$\boldsymbol{\alpha}\big(v_i; z_k\big) = \boldsymbol{f}\Big(v_i; \theta^*\big(\mathcal{D}_{train}\big)\Big) - \boldsymbol{f}\Big(v_i; \theta^*\big(\mathcal{D}_{train}\backslash z_k\big)\Big) \tag{2}$$

For a classification task, the utility function $\boldsymbol{f}(z;\theta)$ for a sample $z = (x,y)$, [32], is defined as:

$$\boldsymbol{f}(z;\theta) = \log\left(\frac{p(z;\theta)}{1 - p(z;\theta)}\right), \tag{3}$$

where $p(z;\theta)$ represents the probability assigned to the correct class by the softmax function of a neural network parameterized by $\theta$. A high $\boldsymbol{f}(z;\theta)$ corresponds to a high likelihood for a given sample $(z)$.

The NTK-based methods like Trak, have a closed-form formulation for data attribution score $(\alpha)$ (Definition 1) expressed as:

$$\boldsymbol{\alpha}(v_j, z_i) = \frac{1}{N} \sum_{n=1}^{N} \left( \boldsymbol{\phi}_n(v_j)^\top (\Phi_n^\top \Phi_n)^{-1} \boldsymbol{\phi}_n(z_i) \right)$$
$$\times \frac{1}{N} \sum_{n=1}^{N} \left( 1 - p_n^{z_i} \right) \tag{4}$$

where, for $N$ different checkpoints of model $(\theta^*)$, $\{p_n^{z_i}\}$ represents the probability assigned by $n^{th}$ set of parameters to the correct class $(y_i)$, for sample $z_i = (x_i, y_i)$. The terms $\phi_n(v_j)$ and $\phi_n(z_i)$ denote the projected gradients of the validation sample $v_j$ and the training sample $z_i$ with respect to the $n^{th}$ set of optimal parameters, and for the utility function $\big(\boldsymbol{f}(\cdot; \theta_n^*)\big)$. Additionally, $\Phi_n$ is the projected gradient for the entire training dataset. Further details about Trak can be found in Appendix B.

To quantify the impact of removing a data sample $z$ from the training dataset on the performance of the entire validation dataset, we define the metric $\mathcal{A}(z)$ as a detrimental attribution score associated with the validation dataset for sample $z$. This metric measures the change in the model's performance $\big(\boldsymbol{f}\big)$ for the validation dataset when $z$ is excluded from the training dataset.

$$\mathcal{A}(z_i) = - \sum_{v_j \in \mathcal{D}_{\text{val}}} \boldsymbol{\alpha}(v_j, z_i)$$
$$= \sum_{v_j \in \mathcal{D}_{\text{val}}} \left( \boldsymbol{f}(v_j; \theta^*(\mathcal{D}_{\text{train}} \setminus z_i)) - \boldsymbol{f}(v_j; \theta^*(\mathcal{D}_{\text{train}})) \right) \tag{5}$$

where $z_i \in \mathcal{D}_{\text{train}}$. Unlike the data attribution score defined in Definition 1, $\mathcal{A}(z_i)$ is the negative of the general definition and evaluates the contribution of each training sample to the likelihood of the entire validation dataset. A higher value of $\mathcal{A}(z_i)$ indicates that removing the training sample $z_i$ and retraining the model with the updated dataset leads to an optimal parameter $\theta^*$ that improves the likelihood of the validation dataset (Equation 3). In other words, training examples that degrade overall validation performance are assigned higher $\mathcal{A}(z_i)$ values. Once $\mathcal{A}(z_i)$ is calculated, it is normalized and used for further steps.

While removing samples with high $\mathcal{A}(z)$ values can improve the model's performance; however, its impact on the downstream model is often tied up with its capability to remove samples with spurious features. During training, spurious features present in the dataset can result in gradient starvation [81, 61], a phenomenon that can hamper the learning of predictive features. Under such scenarios, we theoretically show that the detrimental attribution score($\mathcal{A}$) for a data sample containing a spurious feature ($f_1$) can be lower than that of a data sample with predictive features ($f_2$), even when both features are equally represented. Consequently, deletion strategies based solely on high attribution scores may inadvertently remove examples with predictive rather than spurious features (Proposition 1) and can fail to capture the impact of removing data associated with spurious features on the overall generalization.

**Proposition 1** (Under Valuation of Attribution Scores). *Consider a neural network in the neural tangent kernel (NTK) regime, trained using binary cross-entropy loss with two equally informative features, $f_1$ and $f_2$. lets assume that due to learning dynamics $f_1$ becomes dominant and causes gradient starvation of $f_2$ as per Pezeshki et al. [61]. Then, for two training samples $z_i$ and $z_j$ with equal representation of dominating features $f_1$ and $f_2$, respectively. The attribution score for $z_i$ can be systematically undervalued relative to $z_j$. Formally:*

$$\big|\mathcal{A}(z_i)\big| < \big|\mathcal{A}(z_j)\big|$$

*The proof of Proposition 1, along with further details on gradient starvation, is provided in Appendix F. Empirical evidence supporting this phenomenon is presented in Appendix J.*

This limitation of attribution scores motivates the need for a targeted removal strategy that specifically identifies and eliminates training samples sharing similar spurious features and exhibiting high $\mathcal{A}(z)$ scores. In many practical scenarios, the information about spurious features is missing in the data. Although annotating the entire training dataset using VLM-based models is possible, this approach is often excessively time-consuming and practically infeasible, particularly for large-scale datasets [40]. To address this, we adopt a zero-shot approach [82] and leverage textual descriptions of bias and CLIP embeddings to select data samples that are semantically similar to the identified textual descriptions. Specifically, we convert the textual description (Section 3.2) of the potential spurious feature into an embedding $\mathcal{C}_{\text{text}}$. Similarly, we convert all images in the training dataset into their corresponding CLIP embeddings $\mathcal{C}_{\text{image}}^i$ for $i \in 1, \ldots, |\mathcal{D}_{\text{train}}|$. Each training sample $z_i$ is then assigned a score $\boldsymbol{k}_i$, reflecting its semantic similarity to the identified bias as per the given equation :

$$\boldsymbol{k}_i = \exp\left( -\frac{\big(\mathcal{C}_{\text{text}} - \mathcal{C}_{\text{image}}^i\big) M \big(\mathcal{C}_{\text{text}} - \mathcal{C}_{\text{image}}^i\big)^\top}{2} \right),$$
$$\text{where, } M = LL^\top, L \in \mathbb{R}^{D \times t}, t \ll D \tag{6}$$

The text and image features, denoted as $\mathcal{C}_{\text{text}}, \mathcal{C}_{\text{image}}^i \in \mathbb{R}^{1 \times D}$ are represented as row vectors in a D-dimensional space. The matrix $M$ is a positive semi-definite matrix, constructed as the outer product of a low-rank matrix $L$ (rank at most $t$), and can serve as a learnable transformation. Since $M$ defines the distance metric, varying the values of $L$ allows us to generate different similarity measures for comparing data points [37, 38].

We aim to remove data samples that have high $\mathcal{A}$ scores and are semantically aligned with the identified bias. To achieve this, we learn the matrix $L$ [83, 37, 38] by maximizing the weighted $\mathcal{A}$ score for each sample, where higher weights indicate stronger semantic alignment with bias as per Equation 6. To maintain semantic coherence with the bias description, the cumulative score for the dataset is enforced to exceed a threshold $\mathcal{T}$, defined as a fraction ($\beta$) of the total training size ($|\mathcal{D}_{\text{train}}|$). A larger $\mathcal{T}$ emphasizes semantic alignment, while a smaller $\mathcal{T}$ allows for flexibility in

sample selection based on $\mathcal{A}$ scores. The complete optimization objective is described as below :

$$\max_L \sum_{i=1}^{|\mathcal{D}_{\text{train}}|} \left( \frac{k_i}{\sum_j k_j} \right) \mathcal{A}(z_i)$$

$$\text{s.t.} \quad \sum_{i=1}^{|\mathcal{D}_{\text{train}}|} k_i \geq \mathcal{T}, \quad \mathcal{T} = \beta \times |\mathcal{D}_{\text{train}}|. \tag{7}$$

To ensure that the optimization remains tractable, we replace the hard constraint with a soft penalty term [83] in the objective function. Further detail on this is provided in Appendix C.

Once the optimization is complete, a subset of training data with $k_i$ scores greater than the hyperparameter $\gamma$ is selected for removal ($\mathcal{S}^{\text{deter}}$). The model is then retrained with the updated training dataset ($\mathcal{D}_{\text{train}} \backslash \mathcal{S}^{\text{deter}}$) where, $\mathcal{S}^{\text{deter}} = \{z_i \in \mathcal{D}_{\text{train}} \mid k_i > \gamma\}$. A sensitivity analysis of all the hyperparameters, and comparison with only CLIP and only data attribution on overall performance is provided in Appendix Q and Appendix M, respectively.

## 4 Experiments

### 4.1 Setting

We evaluate the performance of our method across various datasets and compare it with existing data attribution techniques, including original training of model with complete dataset (original), Random deletion of data points (Random), Influence Function (IF) [30], TracIN [58], EWC Repair [31], and Trak [32]. The datasets used in our experiments include WaterBirds [84], Animal with attributes (AWA2) [85], German Traffic Sign Recognition Benchmark (GTSRB) [86], CELEBA [87, 43, 88] (Appendix H), CIFAR-10 [89], and ImageNet-100 [90, 91]. Further comparisons with robustness-based methods(groupDRO [17], JTT [16]) and group balancing methods are provided in Appendix I. For datasets such as GTSRB, CIFAR-10, and WaterBirds, we utilized attributes generated by ChatGPT and VLM models. To further assess the impact of metadata availability, we created two variants for the AWA2 datasets. The first variant, AWA2-A, includes class-specific annotations provided by the original datasets. The second variant, AWA2-B, uses attributes generated using ChatGPT and VLM-based annotation techniques (Section 3.2). All Primary experiments were conducted using a ResNet-18 model, which is the base architecture used in NTK-based data attribution methods such as Trak [32] for the image classification task. Additional experiments using alternative architectures and vision transformer models are presented in Appendix N and Appendix O, respectively. We have reported the worst group accuracy and average accuracy based on prior work on spurious features [67, 17, 28]. However, due to the absence of well-defined group structures in many real-world datasets [17], we have compared these datasets on average accuracy and class-level accuracy. All the experiments were conducted on two NVIDIA A6000 GPUs. Further details on training, hyperparameters, and subset size are provided in Appendix H. Algorithm 1 (Appendix) illustrates the overall workflow of our approach. We also report time and memory overheads associated with subset selection in Appendix R and Appendix S, respectively. Sample images from the selected subset $\mathcal{S}^{\text{deter}}$ are shown in Appendix U.

### 4.2 Improvement in Average Accuracy

Table 1 reports the improvement in average accuracy achieved by our method compared to existing baselines. On average, our method outperforms Trak by 1.4%, EWC by 1.6%, TracIN by 1.4%, Influence Functions by 2.0%, and the original full-dataset training baseline by 1.7%. Notably, we observe gains of 1.9%, 2.5%, and 2.4% over Trak on AWA2-B, WaterBirds, and AWA2-A, respectively. The performance improvement highlights the efficiency of our method in removing the detrimental samples associated with spurious features. We further saw a substantial improvement in under represented class as discussed in Section 4.3. Additional experiments on worst-group accuracy and architectural ablations for WaterBirds are provided in Appendix N.

### 4.3 Class Level Improvement after Data Deletion

Table 2 presents class-level accuracy for datasets with more than two classes. As per the results, our method improves the accuracy of a significant number of classes across datasets. For example, in

Table 1: Comparative evaluation of average accuracy of our proposed method (Ours) against baseline approaches across multiple datasets. The results report mean accuracy scores over three independent runs, with the best-performing values highlighted in **bold**. Entries with a gain of more than 1.5% over full-data training are highlighted in orange, while those exceeding 3% are shown in blue.

| Dataset | Original | Random | IF | TracIN | EWC | Trak | Ours |
|---------|----------|--------|------|--------|------|------|------|
| WaterBirds | 0.638 | 0.606 | 0.603 | 0.652 | 0.650 | 0.656 | **0.681** |
| AWA2-A | 0.644 | 0.622 | 0.644 | 0.652 | 0.642 | 0.638 | **0.662** |
| CELEBA | 0.895 | 0.893 | 0.890 | 0.893 | 0.890 | 0.898 | **0.906** |
| GTSRB | 0.969 | 0.966 | 0.973 | 0.971 | 0.975 | 0.971 | **0.980** |
| AWA2-B | 0.644 | 0.622 | 0.644 | 0.652 | 0.642 | 0.638 | **0.657** |
| CIFAR-10 | 0.774 | 0.787 | 0.798 | 0.784 | 0.789 | 0.793 | **0.801** |
| ImageNet-100 | **0.440** | 0.436 | 0.429 | 0.423 | 0.423 | 0.435 | 0.438 |

Table 2: Class-level accuracy improvement(Imp) after data removal across datasets. The table shows the maximum improvement in any class, the number of improved classes, and the mean improvement across them.

| Dataset | Max Imp | # Imp Classes | Mean Imp |
|---------|---------|---------------|----------|
| Awa2-A | 16.27% | 6 / 10 | 11.12% |
| Awa2-B | 29.16% | 4 / 10 | 17.98% |
| CIFAR-10 | 10.39% | 7 / 10 | 5.59% |
| GTSRB | 50.00% | 22 / 43 | 5.69% |
| ImageNet-100 | 36.00 % | 51 / 100 | 10.15% |

Awa2-A, Awa2-B, ImageNet-100, and CIFAR-10, over 40% of the classes show improvement, with some achieving gains as high as 29.16%. Notably, in GTSRB, 22 out of 43 classes benefit, with a maximum per-class improvement of 50%. The improvement in average accuracy highlights that the improvement in underperforming classes is attained without substantially degrading the performance of other classes. Details on worst-class accuracy are provided in Appendix K.

Table 3: Comparison of best average accuracy across different data attribution methods for different spurious attributes. The table reports the mean accuracy across three independent runs. Entries with a gain of more than 1.5% over full-data training are highlighted in orange, while those exceeding 3% are shown in blue.

| Target | Spurious Attribute | Original | Maj.-Rand | Random | IF | EWC | TracIN | Trak | Ours |
|--------|-------------------|----------|-----------|--------|------|------|--------|------|------|
| arched eyebrows | receding hairline | 0.713 | **0.740** | 0.739 | 0.716 | 0.724 | 0.730 | 0.722 | 0.736 |
| attractive | mouth slightly open | 0.628 | 0.627 | 0.668 | 0.640 | 0.633 | 0.631 | 0.658 | **0.673** |
| big nose | male | 0.771 | 0.770 | 0.770 | 0.764 | 0.751 | 0.745 | 0.756 | **0.780** |
| goatee | bushy eyebrows | 0.946 | 0.931 | 0.947 | 0.938 | 0.951 | **0.953** | 0.949 | **0.953** |
| mouth slightly open | smiling | 0.869 | 0.871 | **0.877** | **0.877** | 0.860 | 0.876 | 0.867 | **0.877** |
| mouth slightly open | wearing lipstick | 0.820 | 0.804 | 0.801 | 0.828 | 0.834 | 0.816 | 0.801 | **0.839** |
| narrow eyes | eyeglasses | 0.840 | 0.858 | 0.862 | 0.856 | 0.858 | 0.860 | 0.855 | **0.862** |
| pointy nose | mouth slightly open | 0.690 | **0.714** | 0.676 | 0.689 | 0.695 | 0.709 | 0.694 | 0.698 |
| receding hairline | rosy cheeks | 0.921 | 0.909 | 0.920 | 0.921 | 0.920 | 0.916 | 0.911 | **0.930** |
| male | pointy nose | 0.919 | **0.931** | 0.907 | 0.909 | 0.911 | 0.906 | 0.915 | 0.921 |

## 4.4 Performance across Different Spurious Attributes

To further investigate the impact of spurious features on both worst-group and average performance, we follow the setup of Eyuboglu et al. [43] and select a subset of the CELEBA dataset where the target attribute is strongly correlated with a spurious feature. We compare the average and worst-group performance achieved by our method against other baselines in Table 3 and Table 4. Additionally, considering the benefit of random data deletion in biased dataset [28] we introduce a new baseline, Maj.-Rand, where the subset of data is randomly deleted from the majority group. As shown in the results, our method outperforms other baselines in average accuracy in 7 and worst-group accuracy in 8 out of 10 settings, respectively. Notably, we observe a gain of over 4% in average accuracy for the target attribute attractive, compared to training on the original dataset. Similarly, worst-group accuracy improves by over 15% for attractive, receding hairline, and arched eyebrows, and by more than 5% for big nose, goatee, and male.

Table 4: Comparison of best worst-group accuracy across different data attribution methods for different spurious attributes. The table reports the mean accuracy across three independent runs. Entries with a gain of more than 5% over full-data training are highlighted in green, while those exceeding 15% are shown in violet.

| Target | Spurious Attribute | Original | Maj.-Rand | Random | IF | EWC | TracIN | Trak | Ours |
|---|---|---|---|---|---|---|---|---|---|
| arched eyebrows | receding hairline | 0.187 | 0.314 | 0.113 | 0.247 | 0.262 | 0.196 | 0.099 | **0.354** |
| attractive | mouth slightly open | 0.213 | 0.242 | 0.347 | 0.266 | 0.241 | 0.205 | 0.392 | **0.407** |
| big nose | male | 0.131 | 0.076 | 0.096 | 0.143 | 0.092 | 0.113 | 0.172 | **0.221** |
| goatee | bushy eyebrows | 0.432 | 0.493 | 0.287 | 0.437 | 0.439 | 0.387 | 0.278 | **0.548** |
| mouth slightly open | smiling | 0.524 | 0.415 | **0.552** | 0.418 | 0.441 | 0.487 | 0.433 | 0.489 |
| mouth slightly open | wearing lipstick | 0.555 | 0.471 | 0.557 | 0.598 | 0.594 | 0.549 | 0.486 | **0.612** |
| narrow eyes | eyeglasses | **0.208** | 0.052 | 0.119 | 0.000 | 0.092 | 0.128 | 0.024 | 0.151 |
| pointy nose | mouth slightly open | 0.045 | 0.044 | 0.046 | 0.034 | 0.028 | 0.021 | 0.040 | **0.084** |
| receding hairline | rosy cheeks | 0.121 | 0.228 | 0.131 | 0.179 | 0.241 | 0.254 | 0.201 | **0.296** |
| male | pointy nose | 0.840 | 0.882 | 0.824 | 0.833 | 0.861 | 0.870 | 0.875 | **0.903** |

Table 5: Comparison of best average and best worst-group accuracy between metadata-driven and VLM-guided textual description.

| Target | Sp. Attribute | Meta Data | | VLM | |
|---|---|---|---|---|---|
| | | Avg. Acc. | WG Acc. | Avg. Acc. | WG Acc. |
| bangs | black hair | **0.922** | **0.649** | 0.916 | 0.624 |
| big nose | wearing necklace | **0.787** | **0.347** | 0.776 | 0.236 |
| heavy makeup | straight hair | 0.826 | **0.716** | **0.835** | **0.716** |
| wearing earrings | bags under eyes | **0.798** | **0.281** | 0.791 | 0.214 |

Table 6: Comparison of best average and best worst-group accuracy between our method and D3M across different spurious attributes.

| Target | Spurious Attribute | Ours | | D3M | |
|---|---|---|---|---|---|
| | | Avg. Acc. | WG Acc. | Avg. Acc. | WG Acc. |
| bangs | black hair | **0.922** | **0.649** | 0.920 | 0.627 |
| big nose | wearing necklace | **0.787** | **0.347** | 0.747 | 0.173 |
| heavy makeup | straight hair | **0.826** | **0.716** | 0.821 | 0.654 |
| wearing earrings | bags under eyes | **0.798** | **0.281** | 0.787 | 0.068 |

## 4.5 Ablation between Meta Data and VLM-based Description

Table 5 compares the performance of our method when using metadata versus VLM-generated textual descriptions of the spurious feature. While both strategies show comparable performance in terms of average accuracy, the metadata-driven variant generally achieves higher worst-group accuracy. This shows that a better annotation of underlying bias can help in the targeted removal of detrimental samples. However, even in the absence of such annotation, LLM and VLM-based methods can generate comparative performance.

## 4.6 Comparison with Group Annotation based Subset Selection

Table 6 presents a comparative evaluation between our method, which relies on the textual description of bias, against a technique that can use group annotation of spurious features in the validation dataset. To compare with such a method, we define group structure based on different values of Spurious Attribute and Target, and then use the method proposed by Jain et al. [11] (D3M) for subset selection. As per the result, on average, our method consistently outperforms D3M across both the best average and worst-group accuracy with a gain of 1.5% in best average accuracy and 11.8% in best worst group accuracy without using the explicit group annotation. This highlights the efficiency of the soft comparison scheme of clip features in handling partially visible features and the proposed optimization scheme compared to hard thresholding used in group annotation.

## 5 Conclusion

In this work, we propose a data deletion framework to mitigate the impact of spurious biases in the training dataset and enhance model performance. Our method employs metric learning techniques to target and remove training samples that are semantically aligned with the textual description of identified biases and whose removal, based on attribution scores, does not adversely affect model performance. To the best of our knowledge, this is the first approach to use text-guided data attribution scores to mitigate simplicity bias in models. However, its effectiveness depends on the quality of the textual descriptions used to capture spurious biases, and the current framework is limited to image datasets. In future work, we aim to incorporate a human-in-the-loop framework to better mitigate complex biases and to extend it to NLP tasks.

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

 # A   Algorithm

---

**Algorithm 1** Proposed Method

---

**Require:** Training Dataset ($\mathcal{D}_{\text{train}}$), Validation Dataset ($\mathcal{D}_{\text{valid}}$), Number of Checkpoints ($M$), Rank for Metric Learning ($t$), Min Weight Fraction ($\beta$), Cutoff for Subset Selection ($\gamma$), CLIP Embedding Model ($\mathcal{C}$), Epochs for Classifier Training ($\mathcal{E}$), Optimization Iterations for Metric Learning ($\mathcal{I}$).

1: **## Classifier Training**
2: i=0
3: **for** epoch $\in [0 \dots \mathcal{E}]$ **do**
4:     Train the classifier using $\mathcal{D}_{\text{train}}$.
5:     **if** epoch $\in [\mathcal{E}, \mathcal{E} - 2, \mathcal{E} - 4, \mathcal{E} - 6, \mathcal{E} - 8]$ **then**
6:         Save the checkpoint $\theta_i$.
7:         i+=1
8:     **end if**
9: **end for**
10: Save $N = [\theta_0, \theta_1, \theta_2, \theta_3, \theta_4]$ checkpoints for the calculation of attribution score as per Equation 4.
11: **## Spurious Feature Identification**
12: Generate a list of possible attributes and corresponding values for $\mathcal{D}_{\text{valid}}$ using ChatGPT (Section 3.2).
13: **for** $i \in [1 \dots |\mathcal{D}_{\text{val}}|]$ **do**
14:     Annotate attribute-value pairs for sample $v_i$ using a Llama-based VLM model (Section 3.2).
15: **end for**
16: **## Calculating the Detrimental Attribution Score**
17: **for** $z_i \in \{z_1 \dots z_n\}$ **do**
18:     Calculate the attribution score $\mathcal{A}(z_i)$ using the saved checkpoints (Equations 4 and 5).
19: **end for**
20: Compare the accuracy of each attribute-value pair using Equation 1. Flag an attribute-value pair as spurious if its accuracy exceeds the average dataset accuracy by a threshold $\tau$.
21: Generate a textual representation of flagged attribute-value pairs under the context of the dataset (Appendix H.4).
22: Create a CLIP embedding of the textual representation ($\mathcal{C}_{\text{text}}$).
23: **## Metric Learning**
24: **for** $i \in [1 \dots |\mathcal{D}_{\text{train}}|]$ **do**
25:     Calculate the CLIP image embedding ($\mathcal{C}_{\text{image}}^i$) for each sample $z_i$ in $\mathcal{D}_{\text{train}}$.
26: **end for**
27: **for** $i \in [0 \dots \mathcal{I}]$ **do**
28:     Optimize the loss $\mathcal{L}$ using $\mathcal{C}_{\text{text}}$, $\mathcal{C}_{\text{image}}$, and $\mathcal{A}(z)$ as per Equation 10 to generate the metric **k** using the hyperparameter t, $\beta$.
29: **end for**
30: Use the score **k**, $\gamma$ to identify $\mathcal{S}^{\text{deter}}$ and retrain the model on $\mathcal{D}_{\text{train}} \setminus \mathcal{S}^{\text{deter}}$.

---

 # B   Details on Trak

$$\boldsymbol{\alpha}(v_j, z_i) = \frac{1}{N} \sum_{n=1}^{N} \left( \boldsymbol{\phi}_n(v_j)^\top (\Phi_n^\top \Phi_n)^{-1} \boldsymbol{\phi}_n(z_i) \right) \times \frac{1}{N} \sum_{n=1}^{N} \left( 1 - p_n^{z_i} \right)$$

$$\text{where, } p_n^{z_i} = (1 + \exp(-y_i \boldsymbol{f}(x_i; \theta_n^*)))^{-1}, \boldsymbol{\phi}_n(v_j) = \mathcal{P}^\top \nabla_\theta \boldsymbol{f}(v_j; \theta_n^*),$$

$$\boldsymbol{\phi}_n(z_i) = \mathcal{P}^\top \nabla_\theta \boldsymbol{f}(z_i; \theta_n^*), \; \Phi_n = \left[ \boldsymbol{\phi}_n(z_1)^\top; \dots; \boldsymbol{\phi}_n(z_{|\mathcal{D}_{\text{train}}|})^\top \right]$$

$$\Phi_n \in \mathbb{R}^{m \times k}, \mathcal{P} \sim \mathcal{N}(0, 1)^{p \times k}, \quad k \ll p. \tag{8}$$

Equation 8, illustrates the calculation of the trak score. Scores consist of an average of the data attribution score calculated over multiple checkpoints (N). The terms $\phi_n(v_j)$ and $\phi_n(z_i)$ denote the projected gradients of the validation sample $v_j$ and the training sample $z_i$ for the $n^{th}$ set of parameters and projection matrix $\mathcal{P}$. This projection matrix reduces the dimension of the gradient $\nabla_\theta \boldsymbol{f}(z; \theta_n^*) \in \mathbb{R}^p$ to a lower-dimensional space $\mathbb{R}^k$, where $k \ll p$, while approximately preserving the inner product, as per the classical Johnson-Lindenstrauss theorem [92].

## C  Soft Penalty for Optimization

For efficient optimisation of the constrained objective presented in Equation 7, we have replaced the hard constraint with a soft constraint $\big(\,d(k)\,\big)$ as per d'Eon et al. [83].

$$d(k) = C \cdot \max\left(\frac{\left(\sum_{i=1}^{|\mathcal{D}_{\text{train}}|} k_i - (\mathcal{T} + w)\right)^2}{w^2}, 0\right), \tag{9}$$

This penalty term is quadratic and scaled by a shrinkable weight w, which is gradually reduced throughout the optimization process. The overall unconstrained optimization problem is defined in Equation 10 where C is a hyperparameter.

$$\mathcal{L} = \max_{L} \sum_{i=1}^{|\mathcal{D}_{\text{train}}|} \left(\frac{k_i}{\sum_j k_j}\right) \mathcal{A}(z_i) - d(k). \tag{10}$$

## D  Notations

Table 7: Notation table for key equation in main draft and proof

| Symbol | Description |
|---|---|
| **General Definitions** | |
| $h(x)$ | Model prediction for input $x$ |
| $y$ | True label corresponding to input $x$ |
| $\mathbf{1}(h(x) = y)$ | Indicator function: 1 if prediction is correct, else 0 |
| $\mathcal{D}_{\text{val}}$ | Validation dataset |
| $\mathcal{D}_{(a^v, b_j^v)}$ | Subset of validation data with attribute–value pair $(a^v, b_j^v)$ |
| $\mathcal{D}_{\text{train}}$ | Training dataset: $\mathcal{D}_{\text{train}} = \{z_1, z_2, \ldots, z_n\}$ where $z_i = (x_i, y_i)$ |
| $\mathcal{X}, \mathcal{Y}$ | Feature set $\mathcal{X} = \{x_1, x_2, \ldots, x_n\}$ and label set $\mathcal{Y} = \{y_1, y_2, \ldots, y_n\}$ |
| $z_k, z_i$ | Training samples from $\mathcal{D}_{\text{train}}$ |
| $v_j$ | Validation sample |
| $\hat{y}$ | Output of the final logit layer of a neural network |
| $\theta$ | Vectorized parameters of the neural network, $\theta \in \mathbb{R}^p$ |
| $e_i$ | Standard unit vectors |
| **Neural Tangent Kernel (NTK) Specific** | |
| $\mathcal{G}(\mathcal{X}, \theta)$ | Neural Tangent Random Feature (NTRF) matrix: $\mathcal{G} = \frac{\partial \hat{y}(\mathcal{X}; \theta)}{\partial \theta}$ |
| $\mathcal{G}_0$ | NTRF matrix at initialization: $\mathcal{G}_0 = \mathcal{G}(\mathcal{X}, \theta_0)$ |
| **SVD Decomposition and Gradient Starvation** | |
| $U, S, V$ | Singular Value Decomposition (SVD) components: $Y\mathcal{G}_0 = USV^\top$ |
| $u^i, v_k$ | Singular vectors from $U$ and $V$ corresponding to features |
| $s_i$ | Singular value representing the strength of the $i^{th}$ feature |
| $\Gamma$ | Response of the network to features: $\Gamma = U^\top Y \hat{y} = SV^T \theta$ |
| $\Gamma_i$ | Response of the $i^{th}$ feature |
| **Attribution and Trak Scoring** | |
| $\Phi_n$ | Stacked gradient features of all training points for model $n$ |
| $\boldsymbol{f}(v; \theta)$ | Model output used for attribution (e.g., logit or loss) for input $v$ under parameters $\theta$ |
| $\boldsymbol{\phi}_n(\cdot)$ | Projected gradient feature under model $n$ |
| $\boldsymbol{\alpha}(v_j, z_i)$ | Attribution score: impact of removing $z_i$ on prediction for $v_j$ |
| $\mathcal{A}(z_i)$ | Detrimental attribution score for training sample $z_i$ |
| $p_n^{z_i}$ | Predicted probability for $z_i$ under model $n$ |
| $\mathcal{P}$ | Random projection matrix with entries drawn from $\mathcal{N}(0, 1)$ |
| **Optimization and others** | |
| $\mathcal{C}_{\text{text}}, \mathcal{C}_{\text{image}}^i$ | Text and image embeddings respectively |
| $\mathcal{T}$ | Trade-off hyperparameter: $\mathcal{T} = \beta \times |\mathcal{D}_{\text{train}}|$ $\big($control tradeoff between data attribution and semantic coherence$\big)$ |
| $\beta$ | Hyperparameter associated with $\mathcal{T}$ |
| $k_i$ | Selection weight for sample $i$ |
| $d(k)$ | Penalty term enforcing deletion constraint |
| $\mathcal{L}$ | Final optimization objective including penalty |
| C | Hyperparameter associated with soft penalty Equation 9 |
| $M = LL^\top$ | Metric matrix constructed from $L$ |
| $L \in \mathbb{R}^{D \times t}$ | Learnable matrix under optimization defined by Equation 7 |
| $\tau$ | Accuracy threshold to detect spurious bias |
| $\theta^*(\cdot)$ | Final model parameters trained on the specified dataset |

## E  Methods for Handling Spurious Features

Table 8 outlines the key capabilities and limitations of existing methods relative to ours. While data augmentation techniques [62–67] are widely adopted, they often require external data, which can conflict with privacy and regulatory constraints [93–96]. Moreover, without appropriate supervision, they risk introducing new spurious features or being vulnerable to data poisoning attacks [97, 98].

In contrast, methods such as group annotation-based optimization (e.g., gDRO [17]), loss reweighting techniques (e.g., JTT [16]), and final-layer fine-tuning [70, 20] do not pose privacy risks. However, in safety-critical applications where models must satisfy stability guarantees [75, 76, 99], these methods can compromise robustness, especially when models are required to ensure Lipschitz continuity for certification. Specifically, they are susceptible to targeted attacks [21–24], particularly when the training procedure heavily relies on a small subset of influential examples [100, 98] used for fine-tuning or reweighting loss values.

Group-balancing techniques [28, 29] partially address these challenges, but often over-prune majority groups. In contrast, our method supports budget-constrained, targeted sample removal, ensuring only detrimental examples are excluded during training.

Furthermore, many of these methods [17, 28, 29] rely on manual group annotations of the training dataset. As spurious features [77] evolve post-deployment, maintaining robustness would require repeated manual annotation cycles. In contrast, our approach eliminates the need for group labels for the training dataset and leverages textual descriptions of bias to guide targeted data removal. The use of a textual description of the bias and the proposed metric learning approach provides a zero-shot approach [101, 82] to approximate the underlying group structure without having any annotation overhead. This design also allows integration of feedback from subject-matter experts, making the process more adaptive and practical.

Table 8: Comparison of methods across regulatory and robustness capabilities.

| Method | Regulatory Restrictions | Supports Textual Descriptions | No Group Annotation in Training Data | Privacy | Prevent Over pruning of Majority Group | Robust to Adv-Attacks |
|---|---|---|---|---|---|---|
| Data Augmentation | ✗ | ✗ | ✓ | ✗ | ✓ | ✗ |
| Group-Annotation based Optimization | ✗ | ✗ | ✗ | ✓ | ✓ | - |
| Reweighting Loss/Data | ✗ | ✗ | ✓ | ✓ | ✓ | ✗ |
| Last Layer Fine-Tuning | ✗ | ✗ | ✓ | ✓ | ✓ | ✗ |
| Group Balancing Method | ✓ | ✗ | ✗ | ✓ | ✗ | ✓ |
| **Ours** | ✓ | ✓ | ✓ | ✓ | ✓ | ✓ |

# F  Theoretical Formulation

For dataset $\mathcal{D}_{\text{train}} = \{z_1 \ldots z_n\}$ where $z_i = (x_i, y_i)$, and $\boldsymbol{x}_i \in \mathbb{R}^d$, and corresponding labels $y_i \in \{-1, +1\}^n$. Let $\hat{y}$ denotes the output of the final logit layer of an L-layer neural network trained using binary cross-entropy, and $\theta \in \mathbb{R}^p$ represents a p-dimensional vectorized parameter of the neural network (Equation 4, Equation 8). let $\mathcal{X} = \{x_1 \ldots x_n\}$ and $\mathcal{Y} = \{y_1 \ldots y_n\}$ constitute the respective features and class labels.

In the Neural Tangent Kernel (NTK) framework [102], the final output of a neural network can be approximated as a linear function of parameters, whose properties are governed by the Neural Tangent Random Feature (NTRF) matrix, defined as:

$$\mathcal{G}(\mathcal{X}, \theta) = \frac{\partial \hat{y}(\mathcal{X}; \theta)}{\partial \theta}, \quad \mathcal{G} \in \mathbb{R}^{n \times p}. \tag{11}$$

For wide-width neural networks, the NTRF matrix remains approximately constant during training [61], allowing the output of the neural network to be approximated using the initial NTRF matrix, $\mathcal{G}_0 = \mathcal{G}(\mathcal{X}, \theta_0)$, as follows:

$$\hat{y}(\mathcal{X}, \theta) = \mathcal{G}_0 \theta. \tag{12}$$

The dominant features of the dataset can be estimated using the principal components of $\mathcal{G}_0 = \mathcal{G}(\mathcal{X}, \theta_0)$, which are equivalent to the principal components of the NTK gram matrix [103].

**Definition 2** (Features and gradient starvation [61]). *Consider a support vector decomposition of $Y\mathcal{G}_0 = USV^\top$, where $Y = diag(y)$, the $i^{th}$ feature is represented by $(V^\top)_{(i,:)}$ or $(V)_{(:,i)}$ with its strength denoted as $s_i = (S)_{ii}$ and its weight across all training samples represented by $(U)_{(:,i)}$. The response of the neural network to the $i^{th}$ feature can be expressed as $\Gamma_i$, where:*

$$\Gamma := U^\top Y \hat{y} = SV^T \theta.$$

*Due to the imbalance in the training dataset, for a given set of features and the optimal parameter $\theta^*$, the presence of the $i^{th}$ feature can influence the learning of the $j^{th}$ feature. This phenomenon, referred to as gradient starvation, arises in optimal parameters if:*

$$\frac{d\Gamma_j^*}{d(s_i^2)} < 0$$

980 Definition 2 suggests that as the strength of the $i^{th}$ feature $(s_i^2)$ increases, the learning of the $j^{th}$ feature gets
981 impacted. This implies that stronger features can dominate the learning process, leading to a reduced contribution
982 of other informative features in the model's predictions.

983 **Theorem 1** (Gradient Starvation Regime [61]). *For a neural network in the linear regime and trained using*
984 *binary cross entropy loss with feature coupling between two features $f_1$ and $f_2$ as defined in Pezeshki et al. [61]*
985 *and with $s_1^2 > s_2^2$, we have,*

$$\frac{d\Gamma_2^*}{d(s_1^2)} < 0,$$

986 Now, under the given setting, we will try to understand the influence of gradient starvation on the performance
987 of the NTK-based data attribution methods :

988 **Proposition 2** (Under Valuation of Trak Scores). *Consider a neural network in the neural tangent kernel (NTK)*
989 *regime, trained using binary cross-entropy loss with two equally informative features, $f_1$ and $f_2$. lets assume*
990 *that due to learning dynamics $f_1$ becomes dominant and cause gradient starvation of $f_2$ as per Pezeshki et al.*
991 *[61]. Then, for two training samples $z_i$ and $z_j$ with equal representation of dominating features $f_1$ and $f_2$*
992 *respectively. The attribution score for $z_i$ can be systematically undervalued relative to $z_j$. Formally:*

$$\left| \mathcal{A}(z_i) \right| < \left| \mathcal{A}(z_j) \right|$$

993 *Proof.* For a sigmoid-based activation, the output probability for feature set $\left( \mathcal{X} \right)$ is given by:

$$p(\mathcal{X};\theta) = \frac{1}{1 + \exp\left( - \hat{y}(\mathcal{X};\theta) \right)},$$
$$p(\mathcal{X};\theta) \cdot \left( 1 + \exp(-\hat{y}(\mathcal{X};\theta)) \right) = 1,$$
$$p(\mathcal{X};\theta) \cdot \exp\left( - \hat{y}(x;\theta) \right) = 1 - p(\mathcal{X};\theta),$$
$$\hat{y}(\mathcal{X};\theta) = \log\left( \frac{p(\mathcal{X};\theta)}{1 - p(\mathcal{X};\theta)} \right).$$

$$(13)$$

994 Hence, the utility function $(\boldsymbol{f})$ used in Trak for data attribution (Equation 3) is equivalent to the logit of a binary
995 cross entropy $(\hat{y})$.

996 From the definition of gradients:

$$\frac{\partial \hat{y}(x;\theta)}{\partial \theta} = \frac{\partial \mathcal{G}_0 \cdot \theta}{\partial \theta} = \mathcal{G}_0 \tag{14}$$

997 As per Equation 4 and Equation 8, $\Phi_m = \mathcal{G}_0 \cdot \mathcal{P}$. Now, considering that the projection matrix [92] preserves
998 the inner product of the actual gradient vector. We will simplify our argument and calculate the value for the
999 unprojected gradients [32] $\left( \mathcal{P} = I_d \right)$. Furthermore, under the NTK regime, where the optimal parameters are
1000 similar [102], we calculate the attribution score for a single checkpoint (M=1). For ease of derivation, we will
1001 omit the subscript m i.e., $\Phi_1 = \Phi$ and $\phi_1 = \phi$, hence:

$$\Phi = \mathcal{G}_0 \tag{15}$$
$$\Phi^T \Phi = \mathcal{G}_0^T \mathcal{G}_0 \tag{16}$$

1002 Now, as per the feature decomposition defined in Definition 2 :

$$Y\mathcal{G}_0 = USV^\top$$
$$\left( Y\mathcal{G}_0 \right)^\top \left( Y\mathcal{G}_0 \right) = \left( USV^\top \right)^\top \left( USV^\top \right)$$
$$\mathcal{G}_0^T Y^\top Y \mathcal{G}_0 = VS^2 V^T$$

$$(17)$$

1003 Since $Y = \mathrm{diag}\{y_1, \ldots, y_n\}$ and $y \in \{-1, 1\}$, it follows that:

$$Y^T Y = I,$$
$$\mathcal{G}_0^T \mathcal{G}_0 = VS^2 V^T,$$
$$\Phi^T \Phi = VS^2 V^T. \tag{18}$$

The validation attribution score (Equation 5) is given by :

$$\mathcal{A}(z_i) = \sum_{v_j \in \mathcal{D}_{val}} -\alpha(v_j; z_i)$$

$$= \sum_{v_j \in \mathcal{D}_{val}} -\phi(v_j)^\top (\Phi^\top \Phi)^{-1} \phi(z_i)(1 - p^{z_i})$$

$$\tag{19}$$

Substituting the value of $\Phi^T \Phi$:

$$\mathcal{A}(z_i) = \sum_{v_j \in \mathcal{D}_{val}} -\phi(v_j)^\top (V S^2 V^T)^{-1} \phi(z_i)(1 - p^{z_i})$$

$$= \left( \sum_{v_j \in \mathcal{D}_{val}} -\phi(v_j)^\top \right) (V)^{-1}{}^\top S^{-2} (V)^{-1} \phi(z_i)(1 - p^{z_i})$$

$$= \left( \sum_{\boldsymbol{v}_j \in \mathcal{D}_{val}} -\phi(v_j)^\top \right) V S^{-2} V^\top \phi(z_i)(1 - p^{z_i})$$

$$= \left( \sum_{\boldsymbol{v}_j \in \mathcal{D}_{val}} -\nabla_\theta f(\boldsymbol{v}_j, \theta)^\top \right) V S^{-2} V^\top \nabla_\theta f(z_i, \theta)(1 - p^{z_i}) \text{ (since } \mathcal{P} = I \text{ and as per Equation 8 )}$$

$$= \left( \sum_{\boldsymbol{v}_j \in \mathcal{D}_{val}} -\nabla_\theta f(\boldsymbol{v}_j, \theta)^\top \right) V S^{-2} V^\top \nabla_\theta f(z_i, \theta)(1 - p^{z_i})$$

$$= \sum_k \frac{\left( (\sum_{\boldsymbol{v}_j \in \mathcal{D}_{val}} -\nabla_\theta f(\boldsymbol{v}_j, \theta)^\top) v_k \right) \left( v_k^\top \nabla_\theta f(z_i, \theta)(1 - p^{z_i}) \right)}{s_{kk}^2}$$

$$\tag{20}$$

where, $v_k$ is the $k^{th}$ column of V matrix and representing the $k^{th}$ feature as per Definition 2

now given the definition of the $\mathcal{G}_0$ and as per Equation 8, Equation 13 and Equation 15

$$\mathcal{G}_0 = [\nabla_\theta f(z_1, \theta)^\top; \ldots; \nabla_\theta f(z_n, \theta)^\top)]$$

$$Y\mathcal{G}_0 = U S V^\top$$

$$\tag{21}$$

For the $i^{th}$ training sample, this score can be further simplified by multiplying with the standard unit vector ($e_i$) on both sides:

$$e_i^\top Y \mathcal{G}_0 = e_i^\top U S V^\top$$

$$y_i \nabla_\theta f(z_i, \theta)^\top = u^i S V^\top$$

$$\tag{22}$$

where $u^i$ is a row vector associated with matrix U,

multiplying both side with $y_i$ and $V$ we get ,

$$y_i \cdot y_i \nabla_\theta f(z_i, \theta)^\top V = y_i u^i S$$

as $y_i^2 = 1$ and further multiplying both side with $e_k$ we get

$$\nabla_\theta f(z_i, \theta)^\top V \cdot e_k = y_i u^i S \cdot e_k$$

$$\nabla_\theta f(z_i, \theta)^\top v_k = y_i u_k^i s_{kk}$$

$$\tag{23}$$

substituting the value in Equation 20 gives :

$$|\mathcal{A}(z_i)| = \left| \sum_k \frac{\left( \sum_{\boldsymbol{v}_j \in \mathcal{D}_{val}} -\nabla_\theta f(\boldsymbol{v}_j, \theta)^\top \right) v_k y_i u_k^i \left( 1 - p^{z_i} \right)}{s_{kk}} \right|$$

$$\tag{24}$$

According to the given equation, for any two data points $z_i$ and $z_j$ where the dominant features are $f_1$ and $f_2$ respectively, the contribution of these features, as per Definition 2, is represented by $u_1^i$ and $u_2^j$. When both dominant features are equally represented, it follows that $u_1^i = u_2^j$ and $u_1^j < u_2^j$, $u_2^i < u_1^i$ . Furthermore, if $|s_{11}| > |s_{22}|$ then as per Theorem 1 $f_1$ induces gradient starvation of $f_2$ and results in lower attribution score i.e., $|\mathcal{A}(z_i)| < |\mathcal{A}(z_j)|$. $\qquad \square$

## G    Data Annotation

### G.1    Attribute Generation

We utilize ChatGPT to generate attributes for a specific dataset with the following prompt referenced from HiBug [39]. The list of attribute-value pairs generated by ChatGPT is provided in Table 9.

*You are a helpful assistant to help user work on improving AI visual models. You need to discuss with your user for a description of the task that the model is working for. You need to decide if the description is complete and clear enough. The description should at least contains or infer the task object, task type, task scene. After understanding user's task description, you should generate related visual attributes that might affect the model's performance. You should not ask me to provide visual attributes. (Note that this is only an example visual attributes according to the previous example, do not take any of its values as default value!): "Gender , Age , Hairstyle , Hair colour" If user is satisfied with the attributes, generate the attribute form with the header formatted as "//Attribute Form//" and end with "//END//". Attributes in the form should be splited by comma. Do not include the task object, task type, task scene. (Note that this is only an example visual attributes according to the previous example, do not take any of its values as default value!):*
*//Attribute Form// Gender , Age , Hairstyle , Hair colour //END//*

Table 9: Details of the attribute value pair generated using ChatGPT.

| Dataset | Attributes | Choices |
|---|---|---|
| AWA2 | Size of the Animal | Small, Medium, Large, Very Large |
| | Fur or Skin Texture of Animals | Smooth, Rough, Furry, Scaly |
| | Color Pattern on Animal | Striped, Spotted, Solid Color, Mixed Colors |
| | Posture of Animal | Sitting, Standing, Flying, Running |
| | Visible Markings or Patterns | Scars, Spots, Unique Patterns |
| | Lighting Conditions | Bright, Dim, Natural, Artificial, Shadowy |
| | Background Complexity | Plain, Cluttered, Natural Habitat |
| | Presence of Humans | None, Nearby, Interacting |
| | Animal Activity State | Resting, Moving, Feeding, Playing |
| | Occlusions | Fully Visible, Partially Hidden |
| | Weather Conditions | Sunny, Cloudy, Rainy, Foggy, Snowy |
| | Seasonal Variations | Summer Coat, Winter Coat, Shedding Fur |
| CELEBA | Gender | Male, Female |
| | Age | Child, Teenager, Adult, Elderly |
| | Facial Expression | Neutral, Smiling, Frowning, Surprised |
| | Hairstyle | Short, Long, Bun, Braided |
| | Hair Color | Black, Brown, Blonde, Red |
| | Skin Tone | Light, Medium, Dark |
| | Facial Hair | Beard, Mustache, Clean-shaven |
| | Presence of Accessories | Glasses, Earrings, Necklace |
| | Lighting Conditions | Bright, Dim, Shadowed |
| | Makeup | Natural, Heavy, None |
| CIFAR-10 | Size | Large, Medium, Small |
| | Pose/Orientation | Side View, Top View, Angled |
| | Lighting | Daylight, Nighttime, Shadows |
| | Background Complexity | Plain, Crowded |
| | Object Occlusion | Partially Visible, Fully Visible |
| GTSRB | Shape of Sign | Round, Triangular, Rectangular |
| | Color of Sign | Red, Blue, Yellow, White |
| | Size of Sign | Small, Medium, Large |
| | Weather Conditions | Sunny, Rainy, Foggy, Overcast |
| | Lighting | Daylight, Nighttime, Shadows, Glare |
| WaterBirds | Surrounding Environment | Forest Floor, Beach, Lake, River, Ocean, Shoreline |
| | Background Elements | Trees, Bushes, Rocks, Water Bodies, Sand, Human-made Structures |
| | Lighting Conditions | Full Daylight, Shaded Areas, Low-light, Overcast |
| | Weather Conditions | Sunny, Cloudy, Rainy, Foggy, Windy |

### G.2    Attribute-Value Annotation

We employ Llama 3.2 [78], a Vision-Language Model (VLM) with 11B parameters, to determine the most suitable value among a set of possible attributes and values for a given dataset. By iterating over a set of images in the validation set, the VLM generates metadata, which is subsequently utilized to identify the spurious features. Each image approximately takes 4-10 seconds on average to annotate, depending on the size of the image. The system prompt provided to Llama 3.2 is as follows:

*You are an expert in identifying visual attributes in a given image. You will be presented with an image along with attributes and a list of choices for each of the attributes. You will be asked to choose the most suited choice for each of the attributes present in the image. Only choose one choice among all given choices*

*for a particular attribute. Ensure that the choice is a string. Reproduce the attribute and the choice as it is.*
*Preserve the case and the spelling. Respond with only a valid JSON object with the attributes as the keys and the*
*chosen choices as the values, and no other extra fluff. Use double inverted commas.*

## H    Training Procedure

### H.1    Model Configuration and Metrics

We maintained consistent hyperparameter settings across all baselines, with the only variation being the subset
of training data selected by each method. The validation set was used to identify underlying spurious biases, as
outlined in Section 3.2. For baseline comparisons, we utilized publicly available implementations. In cases where
the code was not open-sourced or experiments were not conducted on the specific datasets, we implemented
the methods and used the respective datasets for evaluation. For TracIN, we employed the fast implementation
available in the Captum library [104].

Since many real-world datasets lack well-defined group structures [17], which are typically needed for evaluating
worst-group accuracy, we compare our method and baselines primarily on average accuracy. Additionally, to
understand the influence of deleting data samples in mitigating spurious features, we follow the experiment
setup defined by [28, 29, 17] and analyze the worst-case performance improvement. We used the methodology
proposed in [43] to create a subset of CELEBA with specific simplicity biases.

### H.2    Model Training and Datasets

All experiments reported in Table 1 were conducted using the ResNet-18 architecture. The models were trained
from scratch with random initialization. For the WaterBirds dataset, the classifier was trained for 15 epochs
using stochastic gradient descent with a momentum value of 0.9 and a learning rate of 0.001. For all other
datasets, we used the Adam optimizer with a learning rate of 0.001.

The AWA2-A, AWA2-B, CELEBA, models were trained for 15 epochs, while the GTSRB and CIFAR-10 models
were trained for 5 epochs. We have used the same 10 classes as mentioned in Boecking et al. [105] for all
experiments related to AWA2. For CELEBA, we used a subset of 10,000 examples from the original dataset,
with the target label being hair color (blond) and the spurious feature being gender (male). Additionally, we
induced a spurious correlation of 0.4 between the target and spurious features to mimic real-world biases. For
experiments related to ImageNet-100, we have considered the subset of the ImageNet dataset with 100 classes as
per  Tian et al. [91] and trained the model for 10 epochs with the Adam optimizer. We have further considered
the attributes related to texture and shape for common classes available for the ImageNet dataset [106]. The
cutoff value to mark an attribute-value pair as spurious ($\tau$) was decided based on the size of the corresponding
pair in the validation dataset, and the pair generating the largest difference with respect to the original dataset
was picked for analysis.

To ensure a fair comparison for subset selection, we maintained uniformity in the training process across both
the original model training and the retraining process after data deletion.

The experiments reported in Table 3, Table 4 were conducted using the ResNet-18 model, trained for 10 epochs
with the Adam optimizer and a learning rate of 0.001. The dataset was created by randomly sampling the
correlation factor within the range [0,1] and varying the training data size across [5000, 3000, 7000, 10000]. The
correlation attribute and target attribute were selected from the metadata provided in the CELEBA dataset [43].
Experiments on the following target–correlated attribute pairs—(arched eyebrows, receding hairline), (attractive,
mouth slightly open), (big nose, male), (goatee, bushy eyebrows), (mouth slightly open, smiling), (mouth slightly
open, wearing lipstick), (narrow eyes, eyeglasses), (pointy nose, mouth slightly open), (receding hairline, rosy
cheeks), and (male, pointy nose) are conducted with varying training dataset sizes of 3000, 5000, 5000, 5000,
5000, 5000, 7000, 7000, 7000, and 5000 samples respectively, and corresponding spurious correlation strengths
of 0.2, 0.8, 0.4, 0.4, 0.8, 0.9, 0.2, 0.6, 0.6, and 0.6 respectively. Further experiments on the target attributes
Bangs, Big Nose, Heavy Makeup, and Wearing Earrings, were conducted with correlation factors of 0.6, 0.2,
0.4, and 0.2, and with training sample sizes of 10000, 5000, 3000, and 5000, respectively. Results for these
experiments are provided in Table 14

### H.3    Data Attribution

For the experiments reported in Table 1, approximately 3% of the data was removed from the training dataset.
We fix the data removal budget across all baselines, as it is a design choice best left to domain experts. A
smaller removal percentage prevents overpruning of the dataset ( training sample for group land bird on water
is around 56 out of 4795 [29] ) and highlights the precision of attribution methods by focusing on the most
harmful samples. In contrast, larger removals can obscure differences between methods due to overlapping
sample selections. For experiments related to spurious correlation in celeba, considering the stochasticity of the

training sample, we have fixed the budget size to 100 samples. Further ablation on subset size is provided in Appendix Q.1. We ensured uniformity in the data deletion process by basing it on the validation attribution score $\mathcal{A}$, calculated according to the respective definition of data attribution $\alpha$ in each baseline method, using their default hyperparameters.

For our proposed method, we performed hyperparameter tuning by selecting the rank parameter ($t$) from [50, 40, 10, 100] and the minimum weight ($\beta$) from [0.6, 0.7, 0.8, 0.9, 0.95]. The weight barrier ($C$) was chosen from [5, 10]. The optimization for Equation 10 was performed for 5000 iterations using the Adam optimizer with a learning rate of 0.0001. The value of $\gamma$ is decided based on the fraction of the dataset that is removed from the training dataset. For experiments reported in Table 3 and Table 4, hyperparameter tuning was performed over the same range as in previous experiments, optimizing for both best average performance and best worst-group accuracy separately.

## H.4 Textual Description

For different datasets, we used distinct textual representations of the underlying bias. The choice of textual descriptions in our experiments depends not only on the attribute-value pairs but also on the dataset itself. For instance, datasets like AWA2-A contain only label-specific information, such as color and habitat type, without an explicit attribute-value format. Therefore, a suitable textual representation for this dataset could be *"It is a (*1) animal."* Here, (*1) represents the feature identified as a potential biased candidate. Similarly, for GTSRB, incorporating dataset context improves model performance, and a possible template could be *"(*1) of the sign is (2).* where (*1) and (*2) are replaced by the corresponding attribute and value pair.

For datasets such as WaterBirds, AWA2-A, AWA2-B, CELEBA, GTSRB, CIFAR-10, and ImageNet-100 the textual descriptions used in the experiments related to Table 1 are provided in Table 10:

Table 10: Textual Descriptions of Spurious Feature for Different Datasets

| Attribute Description | Dataset |
|---|---|
| *Surrounding environment in image is forest floor* | WaterBirds |
| *It is a domestic animal* | AWA2-A |
| *Size of the animal is very large* | AWA2-B |
| *Image of a male with blond hair* | CELEBA |
| *Shape of the sign is round* | GTSRB |
| *Size of the entity is large* | CIFAR-10 |
| *Object has a spotted pattern* | ImageNet-100 |

For all experiments related to Table 3, Table 4, we used a standardized textual format: *"Image of a person with (*1) and (2)."* where (*1) and (*2) correspond to the target class and the correlated attribute, respectively. Further experiments using VLM-based textual description in Table 5 for the target attributes Wearing Earrings, Bangs, Big Nose, Heavy Makeup use textual description as *"Person is wearing glasses"*, *"Image of a male person"*, *"Person has long hair"*, and *"Person is wearing glasses"* respectively. For metadata, we used the same format as the Table 3.

## I Comparison with Other Optimization and Data-Centric Methods

In general, ImageNet initialization [107] plays a crucial role in achieving strong worst-group accuracy. However, most of our experiments are conducted without ImageNet pretraining to better reflect practical deployment scenarios, particularly those where spurious correlations can significantly degrade model performance [107]. For a fair comparison with optimization-based methods such as gDRO [17] and JTT [16], we additionally evaluate our method on the Waterbirds dataset using a ResNet-18 model pretrained on ImageNet, along with LLM-generated attribute–value annotations. Results averaged over three independent runs are reported in Table 11. We also include comparisons with data deletion methods like D3M [11] and group-balancing approaches such as SUBG and RWG [29].

Table 11 compares the average and worst-group accuracy of our method against various robustness-based approaches on the Waterbirds dataset. Methods are grouped based on whether they require group annotations for the entire training dataset and whether they support textual bias descriptions.

Our method achieves a competitive average accuracy (0.855) and strong worst-group accuracy (0.756) without relying on group annotations, while uniquely supporting textual bias descriptions. Compared to other methods like ERM, D3M, and JTT, our method improves worst-group accuracy by +27.9% over ERM, +12% over JTT,

and +1.6% over D3M. Further comparison with D3M with the same training setup as Table 3 is provided in Table 6.

Group annotation-based methods like gDRO and RWG perform best on worst-group accuracy, but at the cost of requiring explicit group labels for the entire training dataset.

Additional challenges associated with these methods in specific applications are discussed in Section 1, Section 2.2 and Appendix E.

Table 11: Comparison of Average Accuracy and Worst group accuracy achieved by our method in comparison with other robustness-based methods on Waterbirds.

| Method | Group Annotation (Train) | Supports Textual Bias Description | Average Accuracy | Worst Group Accuracy |
|---|---|---|---|---|
| ERM | ✗ | ✗ | 0.819 | 0.477 |
| D3M | ✗ | ✗ | **0.903** | 0.740 |
| JTT | ✗ | ✗ | 0.852 | 0.636 |
| Ours | ✗ | ✓ | 0.855 | **0.756** |
| RWG | ✓ | ✗ | 0.864 | 0.822 |
| SUBG | ✓ | ✗ | 0.833 | 0.814 |
| gDRO | ✓ | ✗ | **0.886** | **0.836** |

## J Empirical Validation of Theoretical Formulation

To validate our theoretical claim, we used the codebase provided by Eyuboglu et al. [43] to sample a 10k subset from CELEBA, where the attributes Male and Smiling are highly correlated. We then computed Trak scores for the training dataset using a ResNet-18 classifier trained to predict the Male label. In this setting, due to the strong correlation between Male and Smiling [43, 108] , smiling may act as a spurious feature. Since the task is to distinguish males from females, we consider features like Beard and Moustache to be more causally relevant, and thus expect that samples with these features to have lower $\mathcal{A}$ scores compared to those with Smiling.

However, statistical analysis of the detrimental attribution($\mathcal{A}$) scores using T-test for the training samples reveals that Smiling has lower scores for samples compared to samples with Beard and Moustache (Table 12). The difference is statistically significant for Beard (p < 0.001). This supports Proposition 1, demonstrating that such effects can arise in practical scenarios.

Table 12: Mean and standard deviation of detrimental attribution ($|\mathcal{A}|$) scores for different attributes, along with statistical significance from a two-sample t-test against *Smiling*.

| Attribute | Mean | Std | p-value (vs Smiling) | Significance |
|---|---|---|---|---|
| Smiling (spurious) | 0.539 | 0.056 | – | – |
| Moustache | 0.545 | 0.044 | 0.1008 | Not significant |
| Beard | 0.544 | 0.036 | 0.00039 | **Significant** |

## K Worst Class Performance

Table 13 reports gains in the worst-performing class for each dataset. In Awa2-A and Awa2-B, worst-class accuracy more than doubles, while in GTSRB, it improves from 50% to 70%. These results demonstrate that our method enhances class-level performance with minimal negative impact on other classes.

Table 13: Worst-class accuracy before and after retraining. The table shows the original worst-class accuracy and the corresponding value after retraining with spurious samples removed.

| Dataset | Original Worst-Class Accuracy | Retrained Worst-Class Accuracy |
|---|---|---|
| Awa2-A | 0.040 | 0.103 |
| Awa2-B | 0.040 | 0.103 |
| CIFAR-10 | 0.589 | 0.575 |
| GTSRB | 0.500 | 0.700 |
| ImageNet-100 | 0.100 | 0.100 |

## L   Group-wise Accuracy Improvements

Table 14: Comparative evaluation of the proposed method (Ours) with the full training baseline (Original) and Trak, reporting the best average and best worst group accuracy (mean$_{std}$) across three runs.

| Target Attribute | Spurious Attribute | Average Accuracy | | | Worst Group Accuracy | | |
|---|---|---|---|---|---|---|---|
| | | Original | Trak | Ours | Original | Trak | Ours |
| Bangs | Black Hair | $0.920_{0.007}$ | $0.921_{0.006}$ | $\mathbf{0.923}_{0.006}$ | $0.523_{0.079}$ | $0.571_{0.053}$ | $\mathbf{0.649}_{0.049}$ |
| Big Nose | Wearing Necklace | $0.765_{0.032}$ | $\mathbf{0.787}_{0.009}$ | $0.787_{0.010}$ | $0.127_{0.065}$ | $0.080_{0.047}$ | $\mathbf{0.347}_{0.148}$ |
| Heavy Makeup | Straight Hair | $0.805_{0.031}$ | $0.800_{0.055}$ | $\mathbf{0.826}_{0.024}$ | $0.651_{0.137}$ | $0.686_{0.078}$ | $\mathbf{0.716}_{0.088}$ |
| Wearing Earrings | Bags Under Eyes | $0.791_{0.020}$ | $0.792_{0.019}$ | $\mathbf{0.798}_{0.028}$ | $0.040_{0.029}$ | $0.017_{0.029}$ | $\mathbf{0.281}_{0.170}$ |

Table 15 presents group-wise accuracy before and after removing samples associated with spurious features associated with Table 14. Groups 1–4 show baseline performance, while Groups 1*–4* report results after pruning. Our method yields notable improvements in some groups without major drops in others.

Table 15: Group-wise accuracy before and after removing spurious samples. The table reports the mean accuracy and standard deviation over 3 runs. Groups 1–4 represent the training with the original dataset, while Groups 1*–4* correspond to results after data pruning.

| Target Attr | Spurious-Attr | G1 | G2 | G3 | G4 | G1* | G2* | G3* | G4* |
|---|---|---|---|---|---|---|---|---|---|
| Bangs | Black Hair | $0.73_{0.07}$ | $0.97_{0.02}$ | $0.53_{0.08}$ | $0.98_{0.01}$ | $0.78_{0.06}$ | $0.96_{0.01}$ | $0.59_{0.12}$ | $0.97_{0.01}$ |
| Big Nose | Necklace | $0.13_{0.06}$ | $0.94_{0.04}$ | $0.32_{0.11}$ | $0.89_{0.07}$ | $0.22_{0.09}$ | $0.94_{0.03}$ | $0.37_{0.11}$ | $0.89_{0.06}$ |
| Heavy Makeup | Straight Hair | $0.68_{0.14}$ | $0.80_{0.12}$ | $0.81_{0.08}$ | $0.82_{0.07}$ | $0.72_{0.11}$ | $0.83_{0.02}$ | $0.80_{0.06}$ | $0.80_{0.02}$ |
| Earrings | Bags Under Eyes | $0.09_{0.04}$ | $0.99_{0.01}$ | $0.04_{0.03}$ | $0.99_{0.01}$ | $0.28_{0.17}$ | $0.96_{0.03}$ | $0.32_{0.16}$ | $0.91_{0.07}$ |

## M   Ablation of Different Components

The ablation study in Table 16 highlights the contribution of key components i.e, data Attribution and CLIP, to the overall performance of our method. For the given experiment, we have used cosine similarity with CLIP (Only CLIP) representation to remove samples that align with the description of the underlying bias. When used independently, both components provide noticeable improvements over the full training baseline, particularly in average accuracy. However, they exhibit limitations in worst-group accuracy when applied in isolation. Notably, combining both Attribution and CLIP in our full method yields the highest performance across nearly all settings, especially in worst-group accuracy, demonstrating the complementary strengths of these components in addressing spurious correlations.

Table 16: Comparative evaluation of the proposed method (Ours) with the full training baseline (Original), Only Attribution, and Only CLIP, reporting the best average and best worst group accuracy (mean$_{std}$) across three runs.

| Target Attribute | Spurious Attribute | Average Accuracy | | | | Worst Group Accuracy | | | |
|---|---|---|---|---|---|---|---|---|---|
| | | Original | Only Attribution | Only CLIP | Ours | Original | Only Attribution | Only CLIP | Ours |
| Bangs | Black Hair | $0.920_{0.007}$ | $0.921_{0.006}$ | $0.922_{0.008}$ | $\mathbf{0.923}_{0.006}$ | $0.523_{0.079}$ | $0.571_{0.053}$ | $0.548_{0.074}$ | $\mathbf{0.649}_{0.049}$ |
| Big Nose | Wearing Necklace | $0.765_{0.032}$ | $\mathbf{0.787}_{0.009}$ | $0.777_{0.002}$ | $0.787_{0.010}$ | $0.127_{0.065}$ | $0.080_{0.047}$ | $0.110_{0.091}$ | $\mathbf{0.347}_{0.148}$ |
| Heavy Makeup | Straight Hair | $0.805_{0.031}$ | $0.800_{0.055}$ | $0.813_{0.010}$ | $\mathbf{0.826}_{0.024}$ | $0.651_{0.137}$ | $0.686_{0.078}$ | $\mathbf{0.739}_{0.045}$ | $0.716_{0.088}$ |
| Wearing Earrings | Bags Under Eyes | $0.791_{0.020}$ | $0.792_{0.019}$ | $0.791_{0.021}$ | $\mathbf{0.798}_{0.028}$ | $0.040_{0.029}$ | $0.017_{0.029}$ | $0.009_{0.013}$ | $\mathbf{0.281}_{0.170}$ |

## N   Architecture-based Ablation on Worst Group Accuracy and Average Accuracy

We further evaluate our method on the WaterBirds dataset across different architectures, including ResNet-18, VGG16, VGG13, AlexNet, and ConvNet. Pham et al. [107] shows that the random initial weights can significantly impact the worst group performance of a model, especially in smaller networks. To replicate this setting, we tested our method under extreme conditions, maintaining consistency in textual instructions and using a single run with the same random seed across all baselines.

As shown in Table 18 and Table 17, In comparison with the complete data setting our method achieves an improvement of 5.0%, 1.9%, 3.6%, 3.1%, 12.6% in worst-group accuracy for VGG16, VGG13, Convnet, ResNet18 and AlexNet architecture and an improvement of 5.2%, 7.1% 4.9% and 7.1% for VGG16, ConvNet, ResNet18, and AlexNet in average accuracy respectively.

Furthermore, compared to Trak, our method achieves an improvement of 1.1%, 2.4%, and 4.8% in average group performance for ConvNet, ResNet18, and AlexNet, respectively. Additionally, enhancements of 5.0%, 1.4%, 7.8%, 4.5%, and 12.9% in worst-group performance were observed for VGG16, VGG13, ConvNet, ResNet18, and AlexNet.

Table 17: Architecture Ablation on WaterBirds (Best Worst Group Accuracy)

| Model | Original | Random | IF | TracIN | EWC | Trak | Ours |
|-------|----------|--------|------|--------|------|------|------|
| VGG16 | 0.053 | 0.050 | 0.064 | 0.064 | 0.062 | 0.053 | **0.103** |
| VGG13 | 0.048 | 0.053 | **0.087** | 0.030 | 0.065 | 0.053 | 0.067 |
| ConvNet | 0.090 | 0.034 | 0.064 | 0.033 | 0.053 | 0.048 | **0.126** |
| ResNet18 | 0.050 | 0.064 | 0.067 | 0.017 | 0.048 | 0.036 | **0.081** |
| AlexNet | 0.050 | 0.048 | 0.107 | 0.031 | 0.042 | 0.047 | **0.176** |

Table 18: Architecture Ablation on WaterBirds (Best Average Accuracy)

| Model | Original | Random | IF | TracIN | EWC | Trak | Ours |
|-------|----------|--------|------|--------|------|------|------|
| VGG16 | 0.640 | 0.669 | 0.657 | 0.640 | 0.683 | 0.686 | **0.692** |
| VGG13 | 0.655 | 0.640 | 0.610 | 0.668 | 0.660 | **0.669** | 0.662 |
| ConvNet | 0.654 | 0.705 | 0.640 | 0.721 | 0.711 | 0.714 | **0.725** |
| ResNet18 | 0.641 | 0.604 | 0.600 | **0.694** | 0.623 | 0.666 | 0.690 |
| AlexNet | 0.644 | 0.650 | 0.586 | 0.693 | 0.658 | 0.667 | **0.715** |

## O  Experiment on Vision Transformer

Existing data attribution methods typically compute gradients over all model parameters, which often causes memory issues for large models like Vision Transformers. To address this, we follow recent works [104, 58] and calculated the gradients only for the final feature layer for both Trak and our method. However, this adaptation was incompatible with other baselines.

The results on Waterbirds for both methods are shown in Table 19.

Table 19: Best Average Accuracy and Best Worst Group performance analysis of our method in comparison with Trak and Original training of vision transformer with entire dataset.

| | Average Accuracy | Worst Group Accuracy |
|--------|------------------|----------------------|
| original | 0.601/0.000 | 0.104/0.000 |
| Trak | **0.644/0.020** | 0.0740/0.014 |
| ours | 0.640/0.027 | **0.1671/ 0.014** |

## P  Relative Comparison with the Baselines

For experiments related to Table 14, we have provided a comparison of the relative performance improvement achieved by our method against other baselines over the complete training data setting. As shown in Table 20 and Table 21, our method, on average, outperforms other baselines in terms of best average accuracy and best worst group accuracy.

Table 20: Relative improvement in Best Average Accuracy (%) achieved by our method and other baselines compared to the complete data setting(Original). The results represent the mean scores from three independent runs, with the best-performing values highlighted in **bold.**

| Target Attribute | Spurious Attribute | Random | EWC | IF | TracIN | Trak | Ours |
|------------------|--------------------|--------|-------|-------|--------|------|------|
| Bangs | Black Hair | -0.03 | 0.37 | 0.36 | -0.07 | 0.16 | **1.05** |
| Big Nose | Wearing Necklace | -0.07 | 1.66 | 1.10 | -0.17 | **2.23** | 2.17 |
| Heavy Makeup | Straight Hair | -0.44 | -1.05 | 0.95 | -2.83 | -0.44 | **2.18** |
| Wearing Earrings | Bags Under Eyes | 0.3 | -0.2 | -1.17 | -0.57 | 0.1 | **0.73** |

Table 21: Relative improvement in Best Worst Group Accuracy (%) achieved by our method and other baselines compared to the complete data setting(Original). The results represent the mean scores from three independent runs, with the best-performing values highlighted in **bold.**

| Target Attribute | Spurious Attribute | Random | EWC | IF | TracIN | Trak | Ours |
|---|---|---|---|---|---|---|---|
| Bangs | Black Hair | 6.25 | **15.62** | 6.66 | 7.05 | 4.77 | 12.58 |
| Big Nose | Wearing Necklace | 4.4 | -3.57 | 5.10 | 4.68 | -4.65 | **22.08** |
| Heavy Makeup | Straight Hair | 1.84 | 5.95 | 4.30 | -1.24 | 3.55 | **6.54** |
| Wearing Earrings | Bags Under Eyes | 5.93 | 1.53 | 13.54 | -3.46 | -3.23 | **24.17** |

## Q  Sensitivity Analysis

Table 22 and Table 23 show the sensitivity of our proposed method on different hyperparameter values.

Table 22: Sensitivity analysis of the average accuracy of our method on the WaterBirds dataset for hyperparameters like the barrier constant ($C$), the matrix rank ($t$) (shown by rows), and the minimum weight fraction ($\beta$, shown by columns).

| Barrier ($C$) | Rank ($t$) | 0.6 | 0.7 | 0.75 | 0.8 | 0.85 | 0.9 |
|---|---|---|---|---|---|---|---|
| | 40 | 0.657 | 0.619 | 0.648 | 0.673 | 0.650 | 0.640 |
| 5 | 50 | 0.673 | 0.642 | 0.621 | 0.618 | 0.618 | 0.601 |
| | 100 | 0.670 | 0.678 | 0.650 | 0.602 | 0.632 | 0.631 |
| | 40 | 0.690 | 0.671 | 0.615 | 0.634 | 0.621 | 0.650 |
| 10 | 50 | 0.633 | 0.679 | 0.639 | 0.657 | 0.629 | 0.609 |
| | 100 | 0.653 | 0.663 | 0.602 | 0.642 | 0.660 | – |

Table 23: Sensitivity analysis of the worst group accuracy of our method on the WaterBirds dataset for hyperparameters like the barrier constant ($C$), the matrix rank ($t$) (shown by rows), and the minimum weight fraction ($\beta$, shown by columns).

| Barrier ($C$) | rank ($t$) | 0.6 | 0.7 | 0.75 | 0.8 | 0.85 | 0.9 |
|---|---|---|---|---|---|---|---|
| | 40 | 0.037 | 0.051 | 0.042 | 0.020 | 0.050 | 0.041 |
| 5 | 50 | 0.033 | 0.042 | 0.055 | 0.048 | 0.056 | 0.065 |
| | 100 | 0.020 | 0.036 | 0.041 | 0.081 | 0.041 | 0.050 |
| | 40 | 0.009 | 0.037 | 0.056 | 0.051 | 0.044 | 0.030 |
| 10 | 50 | 0.044 | 0.023 | 0.053 | 0.031 | 0.051 | 0.061 |
| | 100 | 0.045 | 0.031 | 0.065 | 0.034 | 0.034 | – |

### Q.1  Performance Analysis on Different Subset Size

To further analyze model performance across different subset sizes, we conducted an ablation study where the best hyperparameters were kept fixed while varying the proportion of removed training data. The results are summarized in Table 24.

Table 24: Sensitivity analysis of the worst group accuracy and average accuracy of our method on the WaterBirds dataset for different subset sizes.

| Metrics | 3% | 5% | 15% | 25% |
|---|---|---|---|---|
| Average Accuracy | 0.69 | 0.645 | 0.682 | 0.712 |
| Worst group Accuracy | 0.081 | 0.041 | 0.037 | 0.002 |

## R  Time Taken for Subset Selection

In Figure R, we compare the time taken by our method in comparison with other baselines to select a subset of 1200 images from 60,000 images of CIFAR-10 for the instruction mentioned in Table 10. Since our method uses the attribution scores generated by Trak and improves upon it. The time taken by our method is slightly longer than Trak.

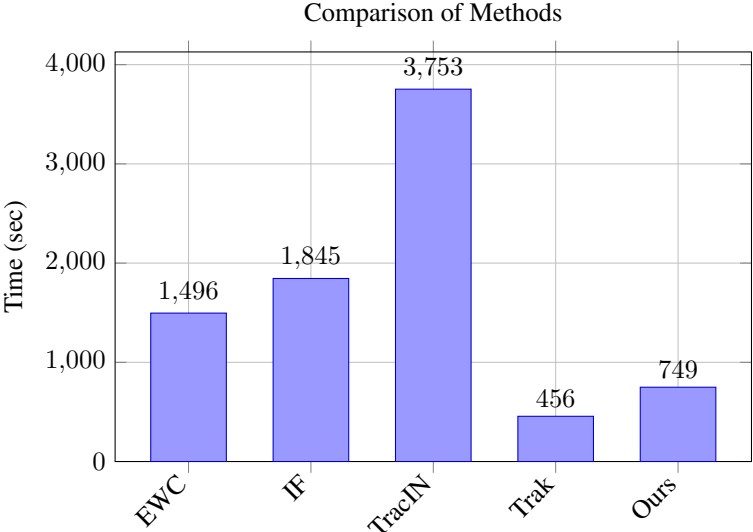

Figure 2: Comparison of time taken to select a subset of 1200 samples from a training dataset of 60,000 images of CIFAR-10 by different baselines and (Ours) for a given textual instruction.

## S  Memory Consumption and Other Training Overhead

The Table 25 reports GPU and RAM usage of our method compared to other baselines, using the same setup described in Appendix R.

As shown, our method introduces only a marginal computational overhead over Trak, which we use for computing data attribution scores. It is to be noted that, while Trak is more memory-intensive, it produces better linear datamodeling score (LDS) scores than other baselines [32].

Table 25: GPU and RAM utilization (in MB) of our method compared to baseline approaches.

| Method | GPU Memory (MB) | RAM Usage (MB) |
|---|---|---|
| IF | 27,749 | 10,578 |
| EWC | 13,221 | 10,520 |
| TracIN | 44,087 | 9,629 |
| **Trak** | 48,020 | 10,710 |
| **Ours** | **48,525** | **10,722** |

 # T    Workflow

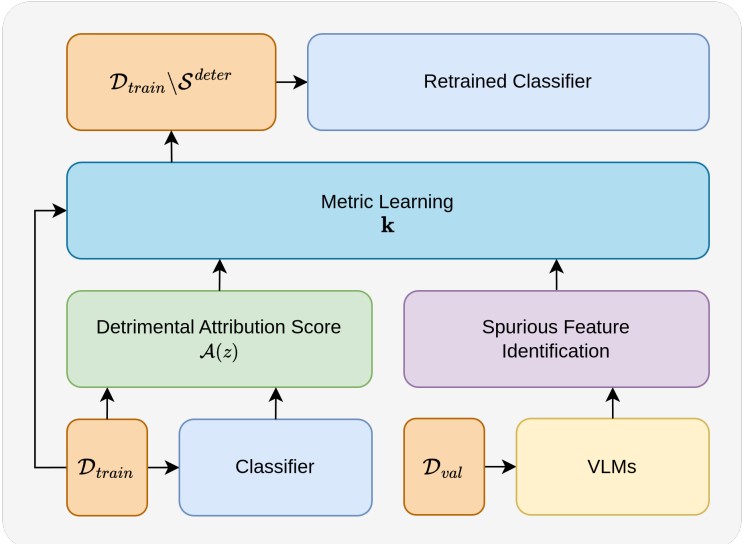

Figure 3: Diagram depicting the workflow of the proposed method

 # U    Images

In this section, we have shown the images that have been removed from the training dataset. Figure 4, Figure 5, Figure 6, and Figure 7 show the set of images that have been removed by our method from the training dataset as ($\mathcal{S}^{\text{deter}}$). For WaterBirds, GTSRB, CELEBA, and AWA2-B, respectively.

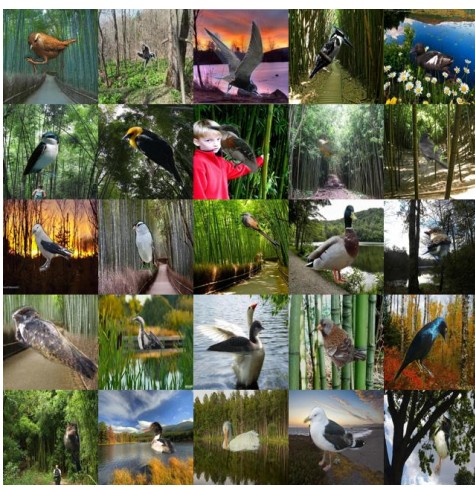

Figure 4: Set of images removed by our method for WaterBirds. The instruction set used for this experiment is *" The surrounding environment in the image is forest floor".*

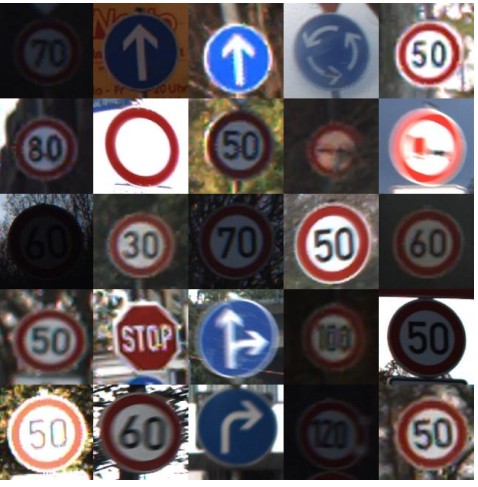

Figure 5: Set of Images removed by our method for GTSRB. The instruction set used for this experiment is *" Shape of sign is round."*

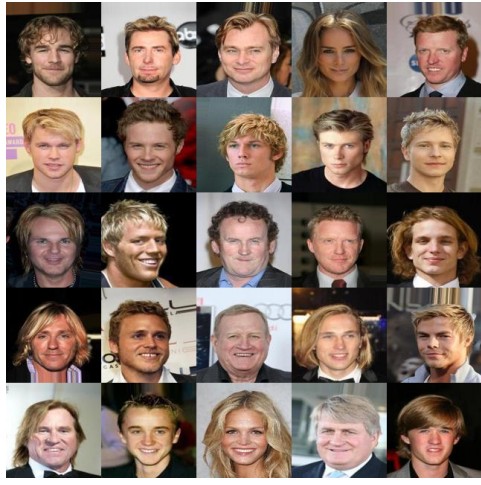

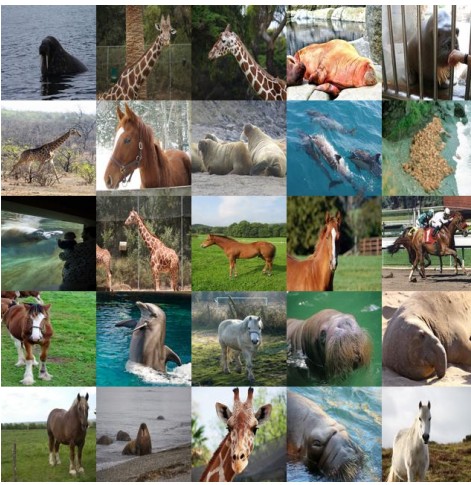

Figure 6: Set of Images removed by our method for CELEBA. The instruction set used for this experiment is *" Image of a male with blond hair"*.

Figure 7: Set of Images removed by our method for Awa2-B. The instruction set used for this experiment is *" The size of the animal is very large."*

