# OpenReview forum: "Text‑Guided Data Attribution: Attributing the Influence of Simplicity Bias to Dataset"
_NeurIPS.cc/2025/Workshop/Reliable_ML — NeurIPS 2025 - Reliable ML Workshop_

### Official Review · Reviewer_VNDz · 2025-09-12
**Good approach, extensive experiments but unclear data usage description**

**Rating:** 4
**Confidence:** 4

**Review:**

The paper presents a novel data attribution framework that utilizes neural tangent kernel and VLM features and predictions. The framework proposes an identification of spurious features using a VLM model and metadata to remove misleading samples from the training set. When metadata is unavailable, the paper proposes a VLM-based annotation. In several benchmarks, the proposed method shows its effectiveness.

**Strengths:**
- The idea is interesting and provides valuable insights for the field of data attribution,
- The paper provides an extensive evaluation of the method.
- The appendix is extensive and provides further Experiments and theoretical insights.
- The paper is clearly written.


**Weaknesses:**
- Clean data usage. The paper discusses a train test split in the introduction and introduces a train validation split in the problem definition. The authors used a VLM to identify spurious features on the validation set. Since no test split is defined in Sec. 3.1 is not clear whether the validation set is used as a test set for the task model and performance measurement. The current description strongly indicates a data leakage, since the train set distribution is directly tailored to the validation/test distribution.
- Given this consideration, it should be reflected in the evaluation if validation/test data without labels is used for metric learning by the methods, and if other methods can also leverage this advantage.
- It is unclear if the method has further modification on $\alpha$ or reuses Trak. The approach used to estimate the attribution score seems pretty straightforward. The attribution score is optimized via metric learning, which is one of the key contributions of this work.

Minor weakness:
- The training epochs with 10 are quite low.
- ChatGPT is not a language model; it is a tool from the GPT family. What is the limitation of the model in the ChatGPT tool?

**Suggestions:**
My main concern is the usage of validation data and the unclarity about how the data is used. Data attribution usually deals with large datasets, and it should be clear if additional data access is required or if the validation set is completely independent from the test set. In addition, an ablation should be conducted if the approaches also work when the training set or a randomly sampled subset of the training set is used for attribution. The data access requirements of the methods should be indicated in the tables in the evaluation.

While the paper has many strengths, a clear description of the data usage and requirements is crucial in my opinion, and a factor for acceptance of the paper.  If this issue can be resolved, I believe this paper has a strong chance of acceptance.

**Ethics:**

I do not have ethical concerns.

---

### Official Review · Reviewer_HjCn · 2025-09-20
**This paper presents a novel text-guided data attribution method to prune spurious samples, though the experimental scope is limited to fully establish its impact.**

**Rating:** 5
**Confidence:** 2

**Review:**

This paper proposes a novel data deletion framework that combines Neural Tangent Kernel (NTK)-based data attribution with textual descriptions of bias to identify and remove training samples that do not significantly affect model performance. They reframe bias mitigation as a data-centric deletion problem guided by natural-language descriptions of spurious features (from metadata or VLMs). They use text as the interface to specify which biases to investigate and prune, instead of requiring full group labels or changing training objectives.

Strengths
- Novel data-centric approach that combines NTK-based data attribution methods with textual descriptions of underlying bias to mitigate the impact of spurious features in training datasets.
- Introduce a metric learning- based data deletion strategy to overcome the limited effectiveness of methods that rely solely on attribution scores for data pruning.
- The evaluation emphasis is on worst-group and class-level gains without explicit group labels.
- They demonstrate improvements across multiple vision benchmarks.

Weaknesses
- The paper focuses on image datasets; claims about broader applicability are not demonstrated.
- Deleting data, even “detrimental” samples, could have downstream effects like distributional shifts or fairness impacts for other subgroups. More rigorous safety/ethical analysis is required to ensure that does not happen.

Ethics - See the second point under Weaknesses. Lacks any discussion on potential impacts on fairness.